# AVHash: Joint Audio-Visual Hashing for Video Retrieval

Yuxiang Zhou
School of Computer Science, BUPT
Beijing, China
yuxiang.zhou@bupt.edu.cn

Zhe Sun
School of Computer Science, BUPT
Beijing, China
2021sunzhe@bupt.edu.cn

Rui Liu
School of Computer Science, BUAA
Beijing, China
lr@buaa.edu.cn

Yong Chen*
School of Computer Science, BUPT
Beijing, China
yong.chen@bupt.edu.cn

Dell Zhang
TeleAI, China Telecom
Shanghai, China
dell.z@ieee.org

## Abstract

Video hashing is a technique of encoding videos into binary vectors, facilitating efficient video storage and high-speed computation. Current approaches to video hashing predominantly utilize sequential frame images to produce semantic binary codes. However, videos encompass not only visual but also audio signals. Therefore, we propose a tri-level Transformer-based audio-visual hashing technique for video retrieval, named AVHash. It first processes audio and visual signals separately using pre-trained AST and ViT large models, and then projects temporal audio and keyframes into a shared latent semantic space using a Transformer encoder. Subsequently, a gated attention mechanism is designed to fuse the paired audio-visual signals in the video, followed by another Transformer encoder leading to the final video representation. The training of this AVHash model is directed by a video-based contrastive loss as well as a semantic alignment regularization term for audio-visual signals. Experimental results show that AVHash significantly outperforms existing video hashing methods in video retrieval tasks. Furthermore, ablation studies reveal that while video hashing based solely on visual signals achieves commendable mAP scores, the incorporation of audio signals can further boost its performance for video retrieval.

## CCS Concepts

• **Information systems** → **Top-k retrieval in databases**; • **Computing methodologies** → **Visual content-based indexing and retrieval**.

## Keywords

Learning to Hash, Video Retrieval, Audio-Visual, Transformer

---

*Yong Chen is the **Corresponding** author at Beijing University of Posts and Communications (BUPT), China. By the way, the first two authors made equal contributions.

---

**ACM Reference Format:**
Yuxiang Zhou, Zhe Sun, Rui Liu, Yong Chen, and Dell Zhang. 2024. AVHash: Joint Audio-Visual Hashing for Video Retrieval. In *Proceedings of the 32nd ACM International Conference on Multimedia (MM '24), October 28-November 1, 2024, Melbourne, VIC, Australia.* ACM, New York, NY, USA, 9 pages. https://doi.org/10.1145/3664647.3681266

## 1 Introduction

In this digital age, short videos[1] have evolved into a prominent content form, emerging as a new narrative style, information source, communication mode, self-expression avenue, and interaction space, significantly influencing modern cognitive psychology, aesthetic experiences, and value preferences [27]. Therefore, the management and processing of videos are crucial yet challenging due to their vast scale and continuous expansion. Video hashing [28], the technique that encodes a video into a binary vector, can not only reduce memory requirements but also facilitate further indexing and computation, making it a potential tool for video analysis and understanding [1, 45].

Current video hashing methods begin by summarizing a given video through a series of keyframes, which are then modeled using non-temporal or temporal networks such as MLP [29, 36], CNN [21, 22], LSTM [7, 16], and Transformer [39]. To name just a few, MCMSH [13] models video feature contexts at three granularities, integrating them into an MLP-Mixer for a comprehensive representation; SRH [12] employs pre-trained CNN features of video keyframes as inputs to an LSTM network for binary code generation; DSVH [2] utilizes 3D convolutions on spatio-temporal keyframes via a 3DCNN to create {0, 1}-embeddings; ConMH [42] leverages a Transformer encoder-decoder on sequential pre-trained CNN features to learn binary codes of videos in a self-supervised fashion. It is evident that the underlying assumption of these approaches is as follows. The current computing capabilities face challenges, or are relatively insufficient, for processing all frames of a video clip, including short videos lasting just a few minutes. However, due to the high redundancy in video frames, summarizing a video's semantic content through keyframe extraction is an effective strategy.

Furthermore, we also observe that existing approaches to video hashing tend to view videos as *Silent Films*, neglecting a crucial element — audio signals — which are prevalent today and vitally important for full video comprehension. Neuroscience research [8] suggests that the combined effect of multiple sensory stimuli can

---

[1]https://digitaldelane.com/the-rise-of-short-form-video-content

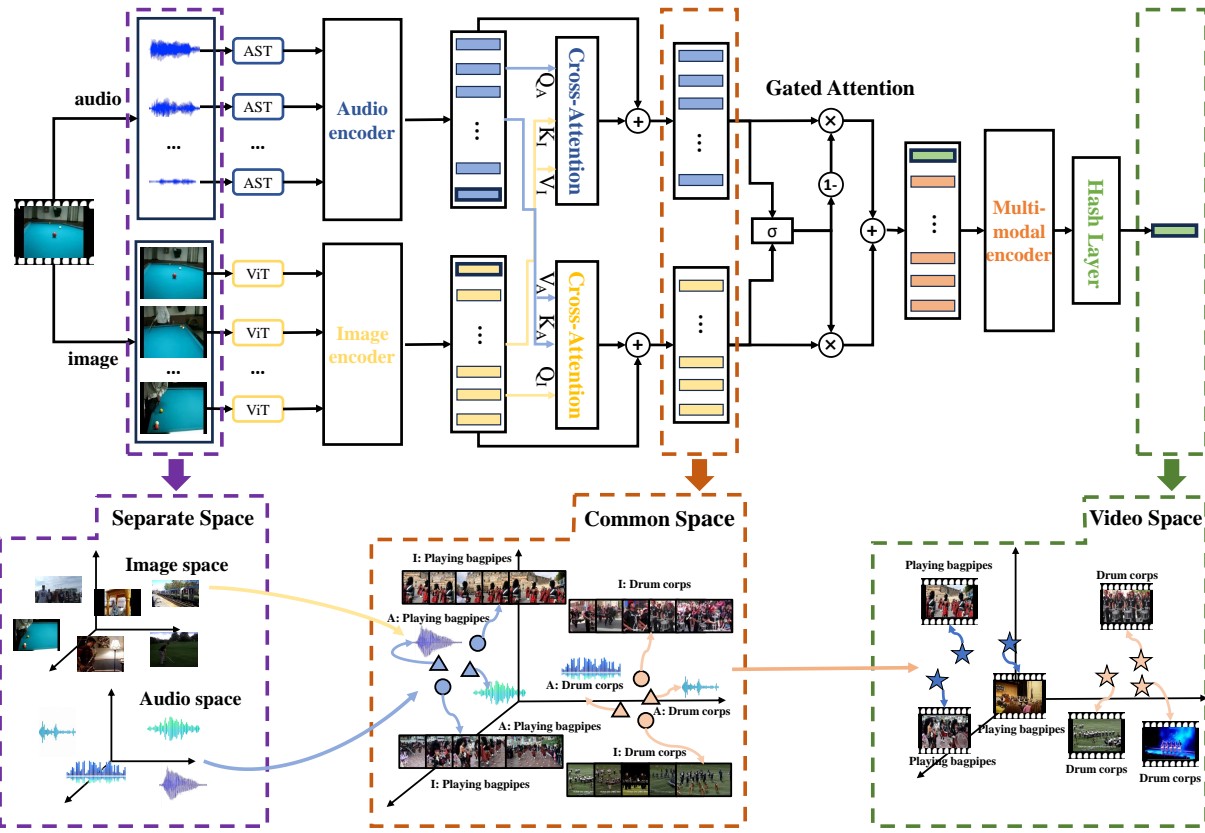

**Figure 1: The architecture of AVHash: from separate spaces to the common latent semantic space, and to the final video space.**

produce an effect where "the whole is greater than the sum of its parts". For example, the concurrent stimulation of sight and sound during the viewing of TV shows or movies can evoke deep and holistic emotions, as well as enhance understanding of the content. In the context of machine learning, the principle that multi-modal learning works better than single-modal has also been mathematically analyzed and empirically validated by Huang et al. [17]. This inspires our investigation into video hashing that integrates both audio and visual signals, instead of solely relying on frame images.

Given a video with synchronized audio-visual signals, we first extract $N$ (e.g., 25) keyframe images and divide its audio into $N$ sound waveform segments; then, we map them to a shared latent space through separate visual-Transformer and audio-Transformer networks respectively; in the end, we combine the audio-visual embeddings into the ultimate video representation using a cross-attention mechanism and an av-Transformer. This completes our method for translating a video from its original signals to its final semantic representation. Given its novel integration of audio and visual signals, we designate this method as "AVHash", whose primary contributions are highlighted below:

- It distinguishes itself by learning to hash videos with both audio and visual components, not merely frame images, and introduces a three-tier Transformer network to map its separate original signals to a shared latent space and finally to the video representation. To the best of our knowledge, this

is the first work that fully leverages both audio and visual elements for video hashing.

- It engages in video-oriented contrastive learning within the final video space and also ensures semantic alignment between audio and visual signals in the common latent space. This approach stems from the understanding that a video's audio-visual signals can be considered not only separately but also collectively as a whole.

- It is validated through comprehensive experiments on two public datasets, ActivityNet and FCVID, under various conditions against a number of state-of-the-art competitors. Notably, AVHash demonstrates significant improvements over those existing methods, showing that while visual-based video hashing already achieves high mAP scores, incorporating audio signals further enhances these metrics.

## 2 Related Work

Video hashing [26, 34, 35, 43, 44, 46] aims to generate a semantic binary code for each video, and therefore significantly reduce storage space and enhance computation speed. This process would be of great benefit to downstream applications including video retrieval, analysis, and understanding.

Current video hashing methods first extract a video's representative image frames and then apply non-temporal or temporal models to create {0, 1} embeddings. The early approaches typically leverage

a pre-trained CNN to extract features from video keyframes first, and then average them to obtain an overall representations. Subsequently, these methods employ image hashing techniques such as LSH [9], ITQ [11], SDH [24, 33], COSDISH [19], SCDH [4], CSQ [47], LTH [3], and MDSH [40] to produce a binary vector, which serves as the video's hash code. More recent approaches predominantly use deep neural networks. For example, DVH [28] utilizes temporal pooling across sequential MLP outputs from video keyframes; SRH [12] captures keyframe CNN features and feeds them into an LSTM network for supervised learning; DSVH [2] extends 2DCNN to 3DCNN for modeling spatio-temporal keyframes; ConMH [42] inputs pre-trained CNN features of sequential keyframes into a Transformer encoder-decoder network, enabling self-supervised learning to hash.

To sum up, existing learning-to-hash methods for video retrieval only utilize visual signals (i.e., frame images) of videos, overlooking a crucial component: audio waveforms. By contrast, our proposed AVHash technique learns to project videos into Hamming space using both visual and audio signals jointly.

In addition, while AVHash utilizes multi-modal signals, it is fundamentally different from existing multi-modal hashing methods such as CLIP4Hashing [49], S²BIN [31], EDMH [5, 6], and SDAH [30]. Those multi-modal hashing techniques aim to learn hash codes for each individual modality (e.g., text, image, or video) for cross-modal retrieval tasks (e.g., image-to-text, text-to-image, video-to-text, text-to-video searches). However, AVHash just learns one-modality's embeddings by fully utilizing its audio-visual components and only performs video-to-video retrieval task. Conversely, AVHash focuses on learning binary embeddings for a single modality, video, while leveraging its audio-visual components, for the purpose of performing video-to-video retrieval better.

## 3 The Proposed Method

### 3.1 Problem Statement

Given a set of $V$ videos, denoted as $\mathcal{S} = \{\mathbf{v}_i\}_{i=1}^{V}$, our AVHash aims to learn a hash function that maps each video to a compact binary code.

Specifically, each video is represented by its $N$ keyframes $\{\mathbf{I}_j\}_{j=1}^{N}$ and $N$ audio segments $\{\mathbf{A}_j\}_{j=1}^{N}$. We extract ViT features [32] $\{\mathbf{f}_j^I\}_{j=1}^{N}$ for keyframes and AST features [10] $\{\mathbf{f}_j^A\}_{j=1}^{N}$ for audio segments, where $I$ and $A$ correspond to the image and audio modalities respectively. These modal features are then fed into their respective branches of the AVHash architecture (see Fig. 1), where the features sequentially pass through separate Transformer encoders to generate image/audio embeddings, a cross-attention for inter-modal interactions, a gated-attention for feature fusion, a multimodal Transformer encoder, and conclude at a hash layer to produce the video's embedding. This embedding is then binarized into the final hash code $\mathbf{b}_i \in \{-1, +1\}^q$ using a Sign($\cdot$) function for video $\mathbf{v}_i$, where $q$ represents the code length.

### 3.2 Model Architecture

Our AVHash consists of the following modules: "Input", " Modality-Specific Encoder", "Cross Attention", "Gated Attention", "Multi-Modal Encoder", "Hash Layer", and "Loss Function" (see Fig. 1).

**Table 1: The configurations of our AVHash architecture.**

| Layer | Configurations |
|---|---|
| 0: Input | Audio Segments: $\{A_1, \cdots, A_N\}$;
Sequential Keyframes: $\{I_1, \cdots, I_N\}$. |
| 1: Feature | Image: CLIP Visual Branch
($L$=24, $H$=1024, $A$=16);
FC layer (1024 × 768);
Audio: AST ($L$=12, $H$=768, $A$=12). |
| 2: Modality-Specific Encoder | Image: ($L_S$=1, $H_S$=768, $A_S$=12);
Audio: ($L_S$=1, $H_S$=768, $A_S$=12). |
| 3: Cross-Attention | I → A: ($L_{I\to A}$=1, $H_{I\to A}$=768, $A_{I\to A}$=12);
A → I: ($L_{A\to I}$=1, $H_{A\to I}$=768, $A_{A\to I}$=12). |
| 4: Gated Attention Fusion Module | FC layer (1536 × 2);
**Tanh**($\cdot$);
Softmax($\cdot$). |
| 5: Multi-Modal Encoder | Transformer Blocks ($L_M$=1, $H_M$=768, $A_M$=12). |
| 6: Hash Layer (Binary-like) | FC layer(768 × $q$);
**Tanh**($\cdot$). |
| 7: Output (Bianry Codes) | **Sign**($\cdot$). |

**Input**. To enable the model to process data from these modalities, we use the AST [10] and CLIP [32] models to extract features from audios and images, respectively, resulting in $\{\mathbf{f}_j^A\}_{j=1}^{N} \in \mathbb{R}^{N \times 768}$ and $\{\mathbf{f}_j^I\}_{j=1}^{N} \in \mathbb{R}^{N \times 768}$. Before feeding them into the subsequent modules, a CLS token needs to be added in front of the feature sequence, i.e.,

$$\mathbf{z}_A^0 = [\text{cls}^A; \mathbf{f}_1^A; \mathbf{f}_2^A; \cdots; \mathbf{f}_N^A] + \mathbf{E}_{\text{pos}}^A, \tag{1}$$

$$\mathbf{z}_I^0 = [\text{cls}^I; \mathbf{f}_1^I; \mathbf{f}_2^I; \cdots; \mathbf{f}_N^I] + \mathbf{E}_{\text{pos}}^I, \tag{2}$$

where $\mathbf{E}_{\text{pos}}^A$ and $\mathbf{E}_{\text{pos}}^I$ correspond to the position encodings of time-series audios and temporal keyframes, respectively.

**Modality-Specific Encoder**. AVHash comprises two modality-specific encoders, designed to capture the unique characteristics inherent to each modality. We employ standard Transformer as audio and visual encoders. Each encoder consists of $L_S$ Transformer layers, and each Transformer layer is mainly composed of Multi-Head Self-Attention (MSA) (with the number of multi-heads as $A_S$) and an MLP layer. The output of $(\ell-1)$-th MLP layer is fed to the $(\ell)$-th Multi-Head Self-Attention (MSA) layer. Layer Normalization (LN) is applied before each layer, and a residual connection is incorporated after each layer. The MLP layer contains two linear fully connected (FC) sub-layers with a non-linear GELU activation function:

$$\mathbf{z}_m^{\ell'} = \text{MSA}(\text{LN}(\mathbf{z}_m^{\ell-1})) + \mathbf{z}_m^{\ell-1}, \tag{3}$$

$$\mathbf{z}_m^{\ell} = \text{MLP}(\text{LN}(\mathbf{z}_m^{\ell'})) + \mathbf{z}_m^{\ell'}, \tag{4}$$

where $\ell = 1, \ldots, L_S$, $m \in \{A, I\}$ denotes the audio or visual modality. By following the above procedure, we could get the latent representation $\mathbf{z}_m^{L_S}$ of video $\mathbf{v}_i$'s audio and keyframe features.

Notice that the output $\mathbf{z}_m^{L_S}$ of each encoder contains $(1 + N)$ parts corresponding to the preset token CLS and $N$ audio/visual features. Then, we use the first part's embedding, dubbed $\mathbf{z}_m^{L_S}[0]$, as the holistic representation and the second part's embedding, dubbed $\mathbf{z}_m^{L_S}[1:]$, as the representation for each feature.

**Cross-Modality Attention**. After unimodal feature encoding, we employ a cross attention to capture the semantic interactions

**Table 2: Dataset Statistics**

| Datasets | #Training | #Validation | #Testing | #Total |
|---|---|---|---|---|
| ActNet-20 | 930 | 237 | 228 | 1,395 |
| ActNet-50 | 2,544 | 666 | 636 | 3,846 |
| ActNet-all | 10,023 | 2,512 | 2,413 | 14,948 |
| FCVID-20 | 3,060 | 1,520 | 1,520 | 6,100 |
| FCVID-50 | 7,993 | 3,973 | 3,983 | 15,899 |
| FCVID-all | 45,508 | 22,754 | 22,754 | 91,016 |

between audio and visual modalities. Concretely, we use a multi-head cross-attention (MCA) with a linear fully connected (FC) layer:

$$\mathbf{z}_{m_2 \to m_1} = \text{FC}(\text{MCA}(\mathbf{z}_{m_1}^{L_S}[1:], \mathbf{z}_{m_2}^{L_S}[1:])) + \mathbf{z}_{m_1}^{L_S}[1:], \quad (5)$$

where $m_1$ denotes the target modality and $m_2$ is the source modality, and the MCA is given as follow:

$$\text{MCA}(\mathbf{z}_{m_1}, \mathbf{z}_{m_2}) = \text{Concat}(\text{head}_1, \dots, \text{head}_{A_c})\mathbf{W}, \quad (6)$$

$$\text{head}_i = \text{Softmax}\left(\frac{\mathbf{Q}_i^{m_1} \times \mathbf{K}_i^{m_2 T}}{\sqrt{d}}\right)\mathbf{V}_i^{m_2}, i = 1, \dots A_c, \quad (7)$$

and

$$\mathbf{Q}_i^m = \mathbf{z}_m \times \mathbf{W}_i^Q, \quad (8)$$

$$\mathbf{K}_i^m = \mathbf{z}_m \times \mathbf{W}_i^K, \quad (9)$$

$$\mathbf{V}_i^m = \mathbf{z}_m \times \mathbf{W}_i^V, \quad (10)$$

where $\mathbf{W}, \mathbf{W}_i^Q, \mathbf{W}_i^K$, and $\mathbf{W}_i^V$ are learnable matrice, and $A_c$ equals to $A_{A \to I}$ or $A_{I \to A}$. Thus, we obtain representations enriched with cross-modal information $\mathbf{z}_{A \to I}, \mathbf{z}_{I \to A}$, by the following formula:

$$\mathbf{z}_{A \to I} = \text{FC}(\text{MCA}(\mathbf{z}_I^{L_S}[1:], \mathbf{z}_A^{L_S}[1:])) + \mathbf{z}_I^{L_S}[1:], \quad (11)$$

$$\mathbf{z}_{I \to A} = \text{FC}(\text{MCA}(\mathbf{z}_A^{L_S}[1:], \mathbf{z}_I^{L_S}[1:])) + \mathbf{z}_A^{L_S}[1:]. \quad (12)$$

**Gated-Attention Fusion Module**. To fuse semantics of different modalities, we introduce a gated attention to determine the importance of each modality. For simplicity, we implement this module via a linear layer followed by a Tanh activation function. This module accepts two modalities' features as input and concatenates the features of each modality by rows. The output of the module is fed into the Softmax function to ensure that the sum of weights across two modalities equals to 1:

$$[\alpha, 1 - \alpha] = \text{Softmax}(\tanh(\text{FC}([\mathbf{z}_{I \to A}, \mathbf{z}_{A \to I}]))); \quad (13)$$

the final fused features are then formulated as:

$$\mathbf{z}_{AI} = \alpha \cdot \mathbf{z}_{I \to A} + (1 - \alpha) \cdot \mathbf{z}_{A \to I}. \quad (14)$$

**Table 3: The number of parameters of different methods.**

| Methods | #Parameters | Methods | #Parameters |
|---|---|---|---|
| MCMSH [13] | 1.2M | DSVH [2] | 63M |
| BTH [25] | 2M | SRH [12] | 4.5M |
| DKPH [23] | 4M | AVH [41] | 22M |
| SSTH [48] | 5M | AVHash (Ours) | 28.4M |
| ConMH [42] | 11.3M | — | — |

**Multi-Modal Encoder**. After fusing the two modality features, to obtain the final representation of the video, we introduce a multimodal Transformer encoder. This encoder consist of $L_M$ Transformer layers, with each composed of Multi-head Self-Attention (MSA) (the number of multi-heads is $A_M$) and an MLP layer. Before feeding $\mathbf{z}_{AI}$, a **cls** token is added at the beginning of $\mathbf{z}_{AI}$, and its corresponding output serves as the video's final representation:

$$\mathbf{z}_V^0 = [\text{cls}^{AI}; \mathbf{z}_{AI}] + \mathbf{E}_{\text{pos}}^{AI}, \quad (15)$$

$$\mathbf{z}_V^{\ell'} = \text{MSA}(\text{LN}(\mathbf{z}_V^{\ell-1})) + \mathbf{z}_V^{\ell-1}, \quad (16)$$

$$\mathbf{z}_V^{\ell} = \text{MLP}(\text{LN}(\mathbf{z}_V^{\ell'})) + \mathbf{z}_V^{\ell'}, \quad (17)$$

where $\ell = 1, \dots, L_M$. By following the above procedure, we can get the latent representation $\mathbf{z}_V^{L_M}$ of video $\mathbf{v}_i$. Note that the output $\mathbf{z}_V^{L_M}$ of each encoder also contains $(1 + N)$ parts corresponding to the preset token CLS and $N$ multimodal features. We employ the first part's embedding, i.e., $\mathbf{z}_V^{L_M}[0]$, as its final representation.

**Hash Layer**. To obtain the final hash codes of video $\mathbf{v}_i$, we add a hash layer, i.e., a FC network with the Tanh($\cdot$) function, on top of the Multi-Modal Encoder. Formally, given video $\mathbf{v}_i$, we transform its Multi-Modal Encoder's output $\mathbf{z}_V^{L_M}[0]$ into a $q$-dimensional binary-like real-valued vector $\mathbf{g}_i$ via:

$$\mathbf{g}_i = \text{Tanh}(\text{FC}(\mathbf{z}_V^{L_M}[0])) \in \mathbb{R}^q, \quad (18)$$

which is then forwarded to the **Sign**($\cdot$) function for the final binary code $\mathbf{b}_i$, i.e.,

$$\mathbf{b}_i = \text{Sign}(\mathbf{g}_i) \in \{-1, +1\}^q. \quad (19)$$

**Loss Function**. We leverage the InfoNCE loss [14, 37] to train our network. InfoNCE loss consists of three types samples: an anchor sample $\mathbf{a}$, a positive sample $\mathbf{p}$, and $n_o$ negative samples $\{\mathbf{n}_i\}_{i=1}^{n_o}$. The anchor and positive samples belong to the same class, while the anchor and negative samples belong to different classes. This InfoNCE loss maximizes the agreement between positive pairs and the dis-agreement between positive and negative samples, which is formulated as:

$$\mathbf{L}(\mathbf{a}, \mathbf{p}, \{\mathbf{n}_i\}_{i=1}^{n_o}) = -\log \frac{\exp\left(\frac{\text{sim}(\mathbf{a}, \mathbf{p})}{\tau}\right)}{\sum_{i=1}^{n_o} \exp\left(\frac{\text{sim}(\mathbf{a}, \mathbf{n}_i)}{\tau}\right) + \exp\left(\frac{\text{sim}(\mathbf{a}, \mathbf{p})}{\tau}\right)}, \quad (20)$$

where $\text{sim}(\cdot, \cdot)$ denotes the similarity function (e.g., cosine similarity), $\tau$ and $n_o$ are two pre-set hyper-parameters. In our experiments, we build 5 types of positive and negative sample pairs:

$$\begin{aligned} \mathbf{L}_1 &= \mathbf{L}_A(\mathbf{a}^A, \mathbf{p}^A, \{\mathbf{n}_i^A\}_{i=1}^{n_o}) + \mathbf{L}_I(\mathbf{a}^I, \mathbf{p}^I, \{\mathbf{n}_i^I\}_{i=1}^{n_o}) \\ &+ \mathbf{L}_{IA}(\mathbf{a}^I, \mathbf{p}^A, \{\mathbf{n}_i^A\}_{i=1}^{n_o}) + \mathbf{L}_{AI}(\mathbf{a}^A, \mathbf{p}^I, \{\mathbf{n}_i^I\}_{i=1}^{n_o}), \end{aligned} \quad (21)$$

and

$$\mathbf{L}_2 = \mathbf{L}_V(\mathbf{a}^V, \mathbf{p}^V, \{\mathbf{n}_i^V\}_{i=1}^{n_o}), \quad (22)$$

where $\mathbf{L}_1$ is the semantic alignment between audio and visual signals in the common latent semantic space, and $\mathbf{L}_2$ represents the video-based contrastive loss in the video space. Then, the overall objective can be constructed as:

$$\mathbf{L} = \alpha \mathbf{L}_1 + \mathbf{L}_2, \quad (23)$$

where $\alpha$ is hyperparameter. By minimizing the InfoNCE loss, AVHash can learn discriminative hash codes, thereby realizing efficient large-scale video retrievals.

Table 4: Different methods' mAP@100 values on ActivityNet and FCVID datasets with 32 and 64 bits.

| Method/mAP/ Dataset | ActivityNet | | | | | | FCVID | | | | | |
|---|---|---|---|---|---|---|---|---|---|---|---|---|
| | -20 | | -50 | | -all | | -20 | | -50 | | -all | |
| | 32 bits | 64 bits | 32 bits | 64 bits | 32 bits | 64 bits | 32 bits | 64 bits | 32 bits | 64 bits | 32 bits | 64 bits |
| MCMSH [13] | 0.1059 | 0.2149 | 0.1508 | 0.2846 | 0.1488 | 0.2849 | 0.2705 | 0.2963 | 0.3642 | 0.3930 | 0.3442 | 0.3723 |
| BTH [25] | 0.1835 | 0.2283 | 0.1845 | 0.2634 | 0.1845 | 0.2721 | 0.2762 | 0.3458 | 0.3083 | 0.3680 | 0.2987 | 0.3593 |
| DKPH [23] | 0.1668 | 0.2251 | 0.1648 | 0.2252 | 0.1528 | 0.2283 | 0.2937 | 0.3535 | 0.3235 | 0.3820 | 0.3256 | 0.3731 |
| SSTH [48] | 0.1636 | 0.2124 | 0.1765 | 0.2408 | 0.1727 | 0.2395 | 0.2017 | 0.2671 | 0.2428 | 0.3057 | 0.2275 | 0.3037 |
| ConMH [42] | 0.2536 | 0.2973 | 0.2887 | 0.3268 | 0.2974 | 0.3312 | 0.3784 | 0.4020 | 0.4040 | 0.4304 | 0.3899 | 0.4232 |
| DSVH [2] | 0.1463 | 0.2200 | 0.2003 | 0.2903 | 0.2056 | 0.2979 | 0.2205 | 0.2900 | 0.2560 | 0.3373 | 0.2616 | 0.3683 |
| SRH [12] | 0.2659 | 0.3116 | 0.3538 | 0.4574 | 0.3205 | 0.4266 | 0.3511 | 0.4768 | 0.3102 | 0.4708 | 0.3699 | 0.4783 |
| AVH [41] | 0.3320 | 0.4203 | 0.3756 | 0.4621 | 0.3576 | 0.4722 | 0.3273 | 0.4580 | 0.3227 | 0.4745 | 0.3318 | 0.4818 |
| AVHash (Ours) | **0.9559** | **0.9731** | **0.9392** | **0.9440** | **0.8530** | **0.8676** | **0.9592** | **0.9622** | **0.9507** | **0.9566** | **0.9208** | **0.9213** |

Table 5: The mAP@100 values of AVHash with different components on ActivityNet and FCVID datasets with 32 and 64 bits.

| Method/mAP/ Dataset | ActivityNet | | | | | | FCVID | | | | | |
|---|---|---|---|---|---|---|---|---|---|---|---|---|
| | -20 | | -50 | | -all | | -20 | | -50 | | -all | |
| | 32 bits | 64 bits | 32 bits | 64 bits | 32 bits | 64 bits | 32 bits | 64 bits | 32 bits | 64 bits | 32 bits | 64 bits |
| audio | 0.6546 | 0.6709 | 0.4071 | 0.4135 | 0.2113 | 0.2175 | 0.7819 | 0.7863 | 0.6495 | 0.6551 | 0.3956 | 0.4007 |
| visual | 0.9372 | 0.9474 | 0.9164 | 0.9370 | 0.8209 | 0.8443 | 0.9184 | 0.9246 | 0.9308 | 0.9361 | 0.9038 | 0.9076 |
| audio+visual | **0.9559** | **0.9731** | **0.9392** | **0.9440** | **0.8530** | **0.8676** | **0.9592** | **0.9622** | **0.9507** | **0.9566** | **0.9208** | **0.9213** |

## 3.3 Training and Inference

The specific configurations of AVHash's architecture is provided in Table 1, and AVHash is implemented and trained (from Layer-0 to Layer-6 in Table 1) via pytorch.

Given a new video $\mathbf{v}_{oos}$, we can use the trained AVHash model to compute its real-valued binary-like vector:

$$\mathbf{g}_{oos} = \mathbf{AVHash}(\mathbf{v}_{oos}) \tag{24}$$

first and then binarize it into the final hash code:

$$\mathbf{b}_{oos} = \mathbf{Sign}(\mathbf{g}_{oos}). \tag{25}$$

## 4 Experiments

### 4.1 Datasets

We assess AVHash using two public video datasets: ActivityNet[2] [15] and FCVID[3] [18]. ActivityNet comprises 14,948 videos, which are categorized into 200 classes. From these, we randomly selected 20 and 50 categories to create two smaller datasets, named ActNet-20 and ActNet-50, respectively. The complete dataset is referred to as ActNet-all. Consequently, we have three ActNet datasets for our experiments. Likewise, FCVID includes 91,016 videos classified into 239 categories, from which we formed three variants: FCVID-20, FCVID-50, and FCVID-all.

---

[2]http://activity-net.org/
[3]https://fvl.fudan.edu.cn/dataset/fcvid/list.htm

Each dataset is split into three partitions: training, validation, and testing sets. Their statistics are presented in Table 2.

### 4.2 Metrics and Competitors

We evaluate the retrieval performance of different approaches using the most widely-adopted metric, mean Average Precision at top-$K$ results (mAP@$K$) [5, 12, 41, 42]. For our study, we set $K$ to 100.

When conducting video retrieval experiments, we designate the testing samples as the query set, with the validation and training sets serving as the video database. Additionally, we order the retrieved results by their Hamming distances from the queries and record the performances of different methods with code lengths of 32 as well as 64 bits.

Regarding baseline approaches, we include SSTH [48], BTH [25], DKPH [23], MCMSH [13], ConMH [42], SRH [12], AVH [41], and DSVH [2], chosen for their competitive retrieval performances. The first five methods are unsupervised, while the last three are supervised.

### 4.3 Implementation Details

**Image processing**. Consistent with SSTH [48], BTH [25], and ConMH [42], we uniformly sample 25 frames from each video in the ActivityNet and FCVID datasets. Each image is resized to 336×336×3 and segmented into 14×14 patches. These images are then processed using the visual branch of CLIP, which features a depth of 24 layers, 16 multi-head attentions, and a hidden size

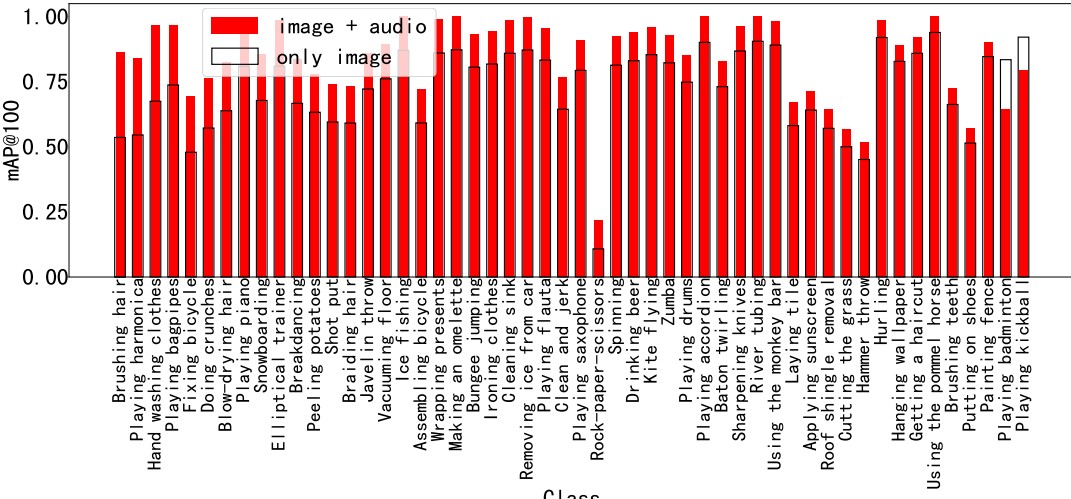

**Figure 2: The mAP@100 values of each class in the ActNet-all dataset (partial classes). The red bars indicate the results of AVHash with audio and visual signals, while the grey bars indicate the results of AVHash only with visual signals.**

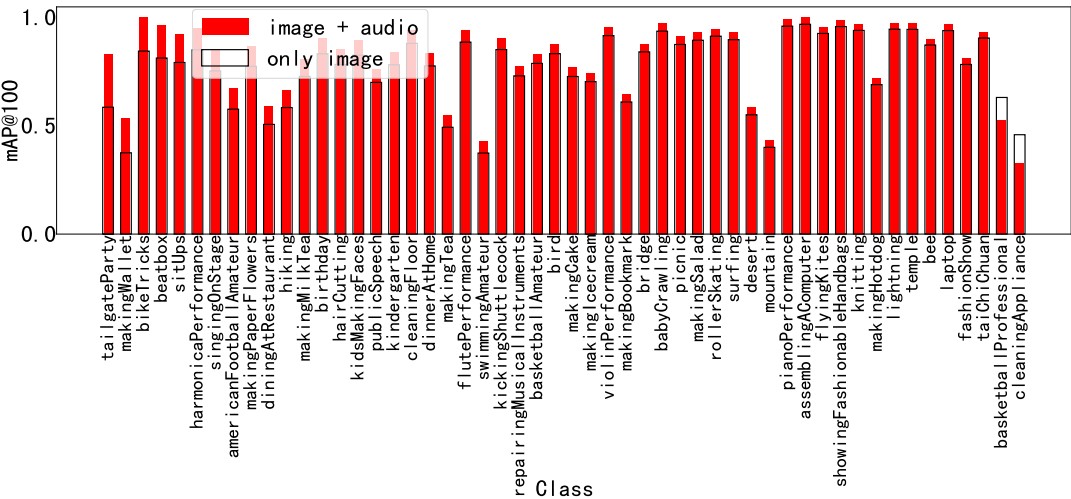

**Figure 3: The mAP@100 values of each class in the FCVID-all dataset (partial classes). The red bars indicate the results of AVHash with audio and visual signals, while the grey bars indicate the results of AVHash only with visual signals.**

of 1024, resulting in high-level image representations within a 1024-dimensional feature space. These features are subsequently transformed to 768-dimensional vectors via a fully connected layer.

**Audio processing**. We also evenly divide each audio track from the ActivityNet and FCVID datasets into 25 segments, corresponding to the 25 frames. These audio segments are then processed using AST [10], which is characterized by an embedding dimension of 768, 12 layers, and 12 heads, generating high-level audio representations within a 768-dimensional feature space.

**Network Architecture**. Our architecture comprises three primary components: the Audio Encoder, the Image Encoder, and the Multi-modal Encoder. Each component employs a Transformer encoder architecture, configured with a single layer, 12 multi-head attentions, and a hidden layer size of 768. Notably, we limit each

component to one layer to ensure that our AVHash network, which has 28.4M learnable parameters, remains comparable in size to competitors such as AVH [41] with 22M parameters and DSVH [2] with 63M parameters, as illustrated in Table 3.

**Model training**. During the training phase, we employ the Adam optimizer [20] for gradient descent, setting the learning rate at $10^{-4}$ and the batch size at 128. In accordance with Eq. (20), we use four negative samples ($n_o$=4) and a temperature parameter ($\tau$) of 0.1. Our AVHash model is implemented using PyTorch on an Nvidia Tesla V100 GPU (32GB). Additional details is available in our code release on Github website: https://github.com/iFamilyi/AVHash. For comparative purposes, the source codes and parameters of competing models are accessible on Github or provided courtesy of their authors.

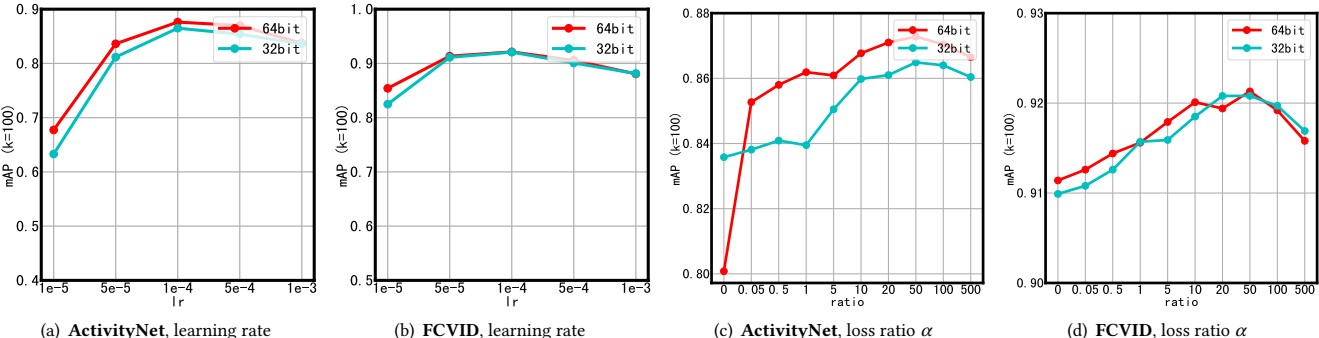

(a) **ActivityNet**, learning rate    (b) **FCVID**, learning rate    (c) **ActivityNet**, loss ratio $\alpha$    (d) **FCVID**, loss ratio $\alpha$

Figure 4: The mAP@100 values w.r.t. different settings of hyper-parameter lr and $\alpha$ on ActNet-all and FCVID-all datasets.

## 4.4 Results

Table 4 presents the mAP@$K$ ($K$=100) values of all the methods in comparison on the ActivityNet (-20, -50, and -all) and FCVID (-20, -50, and -all) datasets. Our AVHash method consistently outperforms the others, including AVH [41], SRH [12], and ConMH [42], across all configurations. Specifically, AVHash achieves impressively high mAP scores — generally exceeding 0.9 — while the scores of competing methods seldom reach 0.5. For instance, at a code length of 32 bits, the mAP@100 scores for AVHash (0.9559 and 0.9731) are approximately 2.88 and 2.32 times higher than those of AVH (0.3320 and 0.4203) on the ActNet-20 and FCVID-20 datasets, respectively. These results strongly suggest the superior efficacy of our proposed approach to video hashing.

One may question why the performance gap between 32-bit and 64-bit configurations is marked for the baseline methods but negligible for our AVHash. This difference can be attributed to the architecture of AVHash (Fig. 1), where the only variation between the 32-bit and 64-bit models is in their final fully connected layer (768×32 vs. 768×64), with both configurations having a similar scale of learnable parameters.

## 4.5 Ablation Studies

We would like to explore the impact of visual and audio signals on the performance of AVHash in video retrieval tasks. This curiosity leads us to conduct comparative experiments: AVHash (with both signals) vs. AVHash with visual input only vs. AVHash with audio input only, denoted as "audio+visual", "visual", and "audio" respectively, as detailed in Table 5. The findings reveal a hierarchy: "audio+visual" outperforms the others, followed by "visual", and "audio" ranks last. It is noteworthy that while visual signals alone could enable AVHash to achieve commendably high mAP@100 scores, the addition of audio signals would further improve AVHash's performance, which affirmatively highlights the positive contribution of audio.

Moreover, we selectively analyze subsets of ActNet-all and FCVID-all, illustrating the mAP@100 values for AVHash under both "audio+visual" and "visual" settings in Fig. 2&3. These figures demonstrate that audio signals generally enhance video comprehension, as indicated by the predominance of red bars over white ones. However, there are instances where audio waveforms might confuse the

interpretation of frame images, suggesting an opportunity for the development of more sophisticated models.

## 4.6 Parameter Sensitivity

Two critical parameters, the learning rate $lr$ and regularization factor $\alpha$ (Eq. (23)), significantly influence our model's performance. Fig. 4 compiles and illustrates their impacts upon mAP@100 scores. Insights from Fig. 4(a)&4(b) reveal that an $lr$ of 1e-4 enables AVHash to attain high performance on both the ActivityNet and FCVID datasets. Likewise, analysis from Fig. 4(c)&4(d) suggests setting $\alpha$ to 50 for optimal results, regardless of whether the dataset is ActivityNet or FCVID. Consequently, we adopt $lr$=1e-4 and $\alpha$=50 as the default settings for our AVHash method.

## 4.7 Convergence Curves

Fig. 5 displays the mAP@100 scores and loss curves for AVHash on both datasets, utilizing 32-bit and 64-bit codes. Each subplot indicates the iteration number on the x-axis, while the left and right y-axes, in blue and red respectively, denote the mAP@100 scores and normalized values of the objective function[4]. The plots clearly demonstrate that AVHash achieves convergence within 300 iterations, thereby confirming the efficacy of AVHash's training process.

## 4.8 t-SNE Visualizations

We selected 10 classes at random, each with 50 samples from the ActNet-all and FCVID-all datasets, and projected them into a 2D space using t-SNE [38], as illustrated in Fig. 6&7. Notably, our AVHash method clusters these videos more distinctly than both the competitive supervised methods AVH/DSVH and the unsupervised approach MCMSH. This demonstrates AVHash's superior capabilities in representation learning.

## 5 Conclusions

This paper sets itself apart from previous studies on video hashing by utilizing both audio and visual components of videos in a tri-level Transformer architecture, named AVHash, for creating binary embeddings of videos. Specifically, AVHash first maps separate audio and video signals to a shared latent semantic space, before

---

[4]The loss for each iteration is normalized via dividing it by the loss of the first iteration.

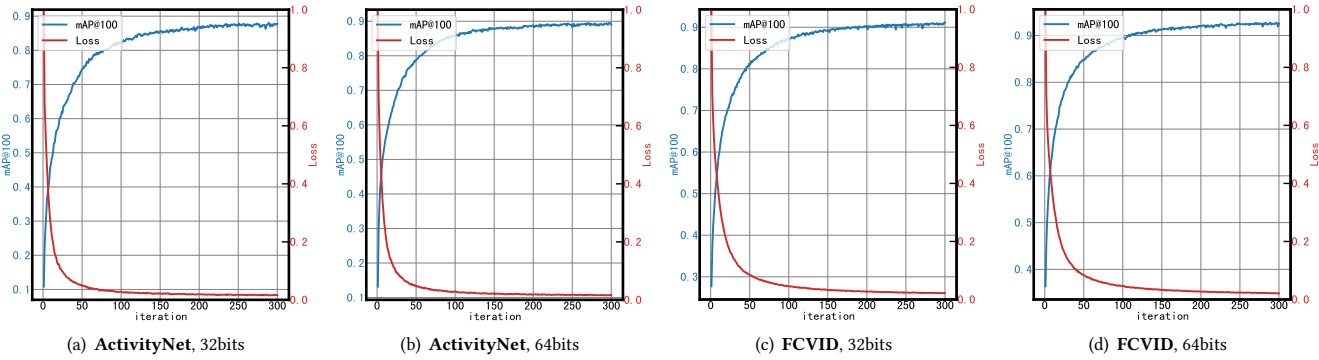

(a) **ActivityNet**, 32bits     (b) **ActivityNet**, 64bits     (c) **FCVID**, 32bits     (d) **FCVID**, 64bits

Figure 5: The mAP@100 and loss curves w.r.t. training iterations on ActNet-all and FCVID-all datasets.

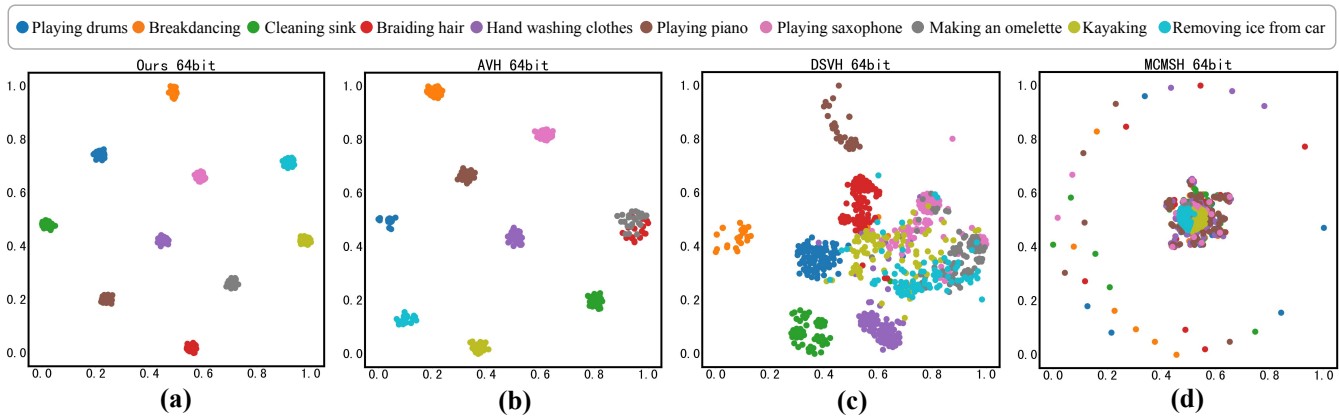

Figure 6: t-SNE visualizations of AVHash and 3 most competitive methods on ActNet-all dataset.

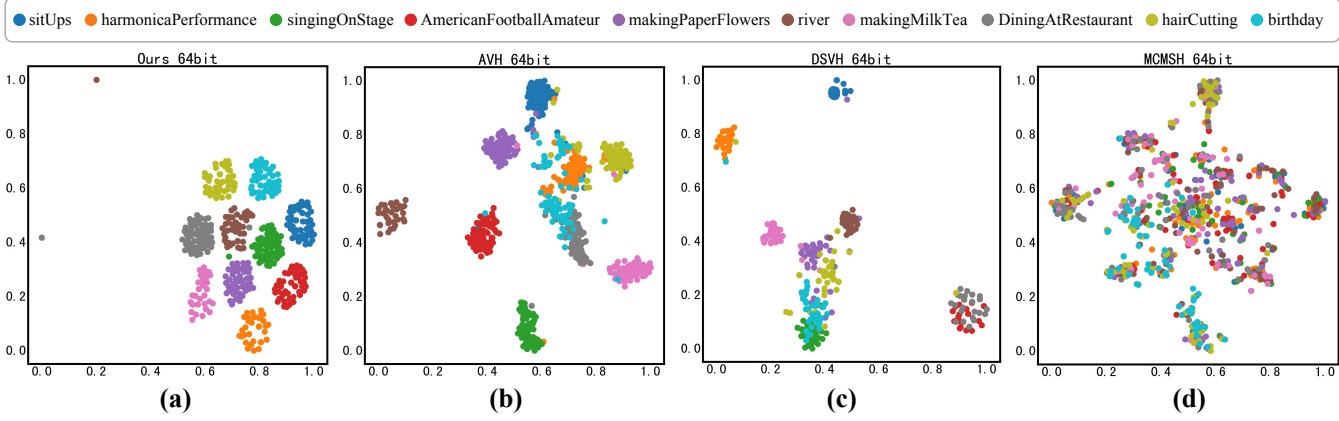

Figure 7: t-SNE visualizations of AVHash and 3 most competitive methods on FCVID-all dataset.

projecting them onto the final video space. It then establishes a contrastive loss in the video space, along with a regularization constraint for aligning audio and visual signals in the common latent semantic space, to guide model training. Extensive experiments conducted on two widely used large video datasets demonstrate that AVHash significantly outperforms existing video hashing techniques in video retrieval tasks. Our findings indicate that, while a high mAP score for video retrieval could be achieved using visual signals alone in video hashing, incorporating audio signals effectively would further improve the system's performance.

# Acknowledgments

This work is supported in part by National Natural Science Foundation of China (NSFC, Grant No. 62372054, 62006005) and National Key Research and Development Program of China (Grant No. 2022YFC3302200). It is also supported in part by Super Computing Platform of Beijing University of Posts and Telecommunications (BUPT).

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

Received 12 April 2024; revised 17 June 2024; accepted 15 July 2024