# OpenReview forum: "AVHash: Joint Audio-Visual Hashing for Video Retrieval"
_acmmm.org/ACMMM/2024/Conference — MM2024 Poster_

### Official Review · Reviewer_YJ8j · 2024-05-18

**Rating:** 2
**Confidence:** 4

**Summary:**

This paper focuses on the video retrieval using hashing. This paper proposes a tri-level Transformer-based audio-visual hashing technique for video retrieval, named AVHash, which leverages audio signals to enhance the learning of video hashing functions. Experimental results show that AVHash outperforms existing video hashing methods in video retrieval tasks.

**Strengths:**

S1:The topic of video hashing is interesting and the use of audio signals for multimodal understanding of video is reasonable.
S2:The overall architecture of the proposed AVHash is feasible.
S3:The introduction of the proposed AVHash is clear.

**Limitations:**

L1:The proposed AVHash is not novel. The architecture (encoding with modality encoder and then fusing with cross-attention/gated attention) is universally filed in multi-modal understanding. This work is just a crude application of an existing method to the hash field.
L2:As far as I am concerned, the introduction of audio signals is not always helpful. For example, in the ActivityNet dataset, much of the audio in the videos does not contribute to action understanding, and much of it is even background noise that can interfere with learning.
L3:The experiment is not comprehensive enough. The authors use only two datasets to verify the effectiveness of the proposed method, which is not convincing.
L4:There seems to be some errors in the experimental settings. It is recommended to use the same settings (dataset splitting and evaluation metrics) with existing baselines such as ConMH and CHAIN. Also, the authors should check the results in Table 4, which I find confusing.
L5:The overall presentation should be improved. In particular, Figures 3 and 4 take up too much space.
L6:The authors should also check the t-SNE figures. It seems that the authors make some mistakes in drawing the t-SNE visualisations (it's a far cry from the results I've seen so often).
L7:The experimental section should be reorganised, e.g. it is suggested to add more analysis to the ablation study and to delete the convergence curves.

**Suitability:**

3

---

### Official Review · Reviewer_z1ho · 2024-05-20

**Rating:** 2
**Confidence:** 4

**Summary:**

This paper introduces AVHash, a three-level Transformer-driven audiovisual hashing system for video search. It takes pre-trained AST and ViT models for separate audio and visual analysis and aligns them into a shared semantic space through Transformer encoding. Training involves audio-visual simultaneous contrastive loss and semantic alignment, allowing AVHash to achieve superior performance in retrieval tasks compared to previous methods.

**Strengths:**

1. The writing of this paper is clear and readers can easily understand the designed modules.
2. The experiment results are very good, much better than existing methods.

**Limitations:**

1. Novelty is limited:

   a) Merging audio signals into video representations is a common practice [1][2][3].You can even integrate more video information, such as subtitles, optical flow and other information.

   b) Network architectures and loss functions for retrieval tasks are common [4].

   c) The authors should focus on innovating the hashing function itself rather than employing well-known engineering techniques.

2. Prior methods are typically tested on datasets like YFCC, which is absent in this paper. This omission might stem from the substantial computational cost associated with model training?
3. Given that the backbone networks employed by the authors are highly advanced, comparisons may be skewed. Would it be feasible to use an identical visual backbone for a fairer comparison?

[1]Audio-based near-duplicate video retrieval with audio similarity learning

[2]Audio-Visual Mismatch-Aware Video Retrieval via Association and Adjustment

[3]Multi-modal Transformer for Video Retrieval

[4]Audio-enhanced text-to-video retrieval using text-conditioned feature alignment

**Suitability:**

3

---

### Official Review · Reviewer_tR9k · 2024-05-30

**Rating:** 5
**Confidence:** 3

**Summary:**

This paper proposes to integrate visual and audio signals to learn hash representations for video retrieval. Experimental results on two datasets demonstrate the effectiveness of the proposed method.

**Strengths:**

The basic idea, jointly learning the visual and audio representation is good.

**Limitations:**

1. The training resources will be a major challenge for jointly learning visual and audio information. I noticed that the authors only conduct the experiments on two datasets, FCVID and ActivityNet. However, there is another commonly used dataset, YFCC, which is far larger than these two. I wonder if the computation resource is enough to train the model on YFCC and how long it will take to train the model?
2. In L364, it seems that the target modality and source modality are not explained. Do they mean visual and audio modalities?
3. Figure 3 and Figure 4 shows the performance with/without audio. It turns out that two categories in each dataset show worse performance when adding audio. I think it would be better to visualize the audio signals of them to analyze when the audio will help and when will not.
4. L729-L732 is not convincing enough. As far as I know, most video hash methods use the final fully connected layer to control the code length. So I think there should be another reason for the phenomenon.
5. Just a minor suggestion, Figure 1 can be improved as it tells little information.
6. Some relevant hashing-based video retrieval works are not included.
e.g. Li et al., Structure-adaptive neighborhood preserving hashing for scalable video search, TCSVT, 2021.

**Suitability:**

3

---

### Official Review · Reviewer_VvDd · 2024-06-01

**Rating:** 5
**Confidence:** 3

**Summary:**

Video hashing constitutes a sophisticated method for converting video data into compact binary sequences, enabling optimized storage solutions and rapid computational tasks. Departing from standard practices that mainly leverage sequential frames for the derivation of semantic binary fingerprints, this paper introduces an innovative methodology through AVHash. This approach is underpinned by a pioneering tri-level Transformer design that harmoniously synthesizes both the auditory and visual facets of videos into binary representations.

**Strengths:**

The paper's key advantages are summarized as follows:
1.Joint Audio-Visual Hashing: Pioneering a novel concept where both audio and visual signals are harnessed together for video hashing, breaking from the norm of relying solely on visual frames.
2.Gated Attention Fusion: Introducing a mechanism for audio-visual signal fusion that selectively weighs and integrates information, enhancing the interaction between modalities.
3.Tri-Level Transformer Network: Implementing a distinctive three-stage Transformer architecture that initially separates, aligns, and then fuses signals in a shared semantic space, enriching representations.
4.Contrastive and Regularized Training: Employing a dual strategy with contrastive loss for video-centric alignment and a regularization term for audio-visual coherence, guiding the model towards optimal retrieval performance.

**Limitations:**

While the AVHash methodology marks a substantial advancement in video retrieval by integrating audio-visual information effectively, there are several shortcomings should be addressed:
1.In section 4.3, although the comparison of 32-bit and 64-bit code length is mentioned, the influence of different code lengths on retrieval accuracy and computational cost, and the basis for selecting the optimal code length were not analyzed in depth.
2.Audio Quality Variation: Real-world videos have variable audio conditions, such as noise, accents, and background sounds. The paper doesn't extensively evaluate robustness under diverse audio variations, potentially affecting practical performance.
3.In figure 3&4, there are some cases that the performance of image + audio is reduced, and the authors do not seem to perform an in-depth analysis.

**Suitability:**

3

---

### Official Review · Reviewer_Mcwo · 2024-06-06

**Rating:** 5
**Confidence:** 3

**Summary:**

This work proposes a video hashing technique that encodes videos into binary vectors using both audio and visual signals. The technique employs a tri-level Transformer-based model that separately processes audio and visual signals using pre-trained AST and ViT large models, then projects these into a shared latent semantic space with a Transformer encoder. A gated attention mechanism fuses the paired audio-visual signals, leading to the final video representation. The model is trained using a video-based contrastive loss and a semantic alignment regularization term for audio-visual signals. Experimental results demonstrate that AVHash significantly outperforms existing video hashing methods in video retrieval tasks.

**Strengths:**

Well written and well structured paper, with clear model architecture explanation. The proposed method demonstrates impressive effectiveness and improvement over baseline models. This approach has many practical applications for efficiency video indexing and retrieval. The t-SNE clustering results imply that the hashes closely cluster similar categories and provides clear separation between groups.

**Limitations:**

- There appears to be a correlation between number of parameters and mAP@100 scores, so perhaps larger models are inherent better. However, 28.4 million parameters may be difficult to deploy on resource-constrained environments.

- Although the the model outperforms all other model on small datasets, its performance diminishes on larger datasets while other models continue to improve. I suspect scalability issues to larger and more diverse datasets.

- According to Table 5, video alone is able to achieve a very impressive mAP@100 score, higher than all other methods. This raises questions about why existing hashing methods that solely use frame images cannot achieve similar performance levels.

**Suitability:**

2

---

### Meta-Review · Program_Chairs · 2024-06-25

**Recommendation:** Accept (Poster)
**Confidence:** 5

**Metareview:**

Two reviewers simultaneously agreed on the limited novelty issue within the submission, the responses may not be convincing to address such major concern.

***TPC Addendum***
The paper has received a mixed verdict. Its limitations in terms of novelty are noted. So are its strengths in terms of jointly learning the visual and audio representation and good presentation. The paper is innately multimodal which is a priority for this conference, hence the TPC is recommending a poster accept to allow for the conversation to continue at the conference.